# Hematopoietic Stem Cell Transplantation in *CSF1R*-Related Leukoencephalopathy: Retrospective Study on Predictors of Outcomes

**DOI:** 10.3390/pharmaceutics14122778

**Published:** 2022-12-12

**Authors:** Jarosław Dulski, Michael G. Heckman, Launia J. White, Kamila Żur-Wyrozumska, Troy C. Lund, Zbigniew K. Wszolek

**Affiliations:** 1Department of Neurology, Mayo Clinic, 4500 San Pablo Rd, Jacksonville, FL 32224, USA; 2Division of Neurological and Psychiatric Nursing, Faculty of Health Sciences, Medical University of Gdansk, 80-211 Gdansk, Poland; 3Neurology Department, St Adalbert Hospital, Copernicus PL Ltd., 80-462 Gdansk, Poland; 4Division of Clinical Trials and Biostatistics, Mayo Clinic, Jacksonville, FL 32224, USA; 5Department of Medical Education, Jagiellonian University Medical College, 30-688 Kraków, Poland; 6Department of Pediatrics, Division of Blood and Marrow Transplant, University of Minnesota, Minneapolis, MN 55455, USA

**Keywords:** adult-onset, axonal spheroids, pigmented glia, ALSP, HSCT, bone marrow transplant, neurodegenerative, white matter, leukodystrophy

## Abstract

Mutations in the *CSF1R* gene are the most common cause of adult-onset leukoencephalopathy with axonal spheroids and pigmented glia (ALSP), a neurodegenerative disease with rapid progression and ominous prognosis. Hematopoietic stem cell transplantation (HSCT) has been increasingly offered to patients with *CSF1R*-ALSP. However, different therapy results were observed, and it was not elucidated which patient should be referred for HSCT. This study aimed to determine predictors of good and bad HSCT outcomes in *CSF1R*-ALSP. We retrospectively analyzed 15 patients, 14 symptomatic and 1 asymptomatic, with *CSF1R*-ALSP that underwent HSCT. Median age of onset was 39 years, and the median age of HSCT was 43 years. Cognitive impairment was the most frequent initial manifestation (43%), followed by gait problems (21%) and neuropsychiatric symptoms (21%). Median post-HSCT follow-up was 26 months. Good outcomes were associated with gait problems as initial (*p* = 0.041) and predominant (*p* = 0.017) manifestation and younger age at HSCT (*p* = 0.044). Cognitive impairment as first manifestation was a predictor of a bad outcome (*p* = 0.016) and worsening of cognition post-HSCT (*p* = 0.025). In conclusion, gait problems indicated a milder phenotype with better response to HSCT and good therapy outcomes. In contrast, patients with a higher burden of cognitive symptoms were most likely not to benefit from HSCT.

## 1. Introduction

Mutations in the *Colony Stimulating Factor 1 Receptor (CSF1R)* gene are the most common cause of adult-onset leukoencephalopathy with axonal spheroids and pigmented glia (ALSP) [1,2]. The disease is characterized by progressive cognitive decline, neuropsychiatric features (anxiety, apathy, depression, disinhibition, irritability, and other frontal lobe symptoms), parkinsonism, pyramidal signs, gait difficulty, and imbalance [1,2]. Some patients may also manifest seizures and stroke-like exacerbations of the disease [1,2]. The prognosis is ominous, with rapid progression and death within a few years from the initial symptoms [1,2].

In recent years, advances in research on *CSF1R*-ALSP have uncovered pathophysiological mechanisms and opened new treatment perspectives for the previously untreatable disease [2,3,4]. The CSF1R protein is primarily expressed in the central nervous system, where it is essential for the normal development, differentiation, function, and maintenance of the microglia [1,2]. *CSF1R* mutations lead to haploinsufficiency, but the remaining microglia are sufficient to compensate during the first few decades of life. However, with aging, there is a growing accumulation of damage to white matter, and once the compensation abilities are overwhelmed, the symptomatic disease ensues [1]. Microglia replacement by hematopoietic stem cell transplantation (HSCT) offered potential means of stabilizing the disease progression and improving survival [4,5]. Since the first report on the application of HSCT in *CSF1R*-ALSP by Eichler and colleagues in 2016 [6], there have been several other case reports and series. Many patients significantly benefited from the transplantation; however, some experienced worsening of their condition or even death. To date, no study has evaluated factors that may be predictive of response to HSCT. In the current study, we attempt to fill that gap in knowledge and assess predictors of good and bad HSCT outcomes in *CSF1R*-ALSP.

## 2. Methodology

We searched the medical literature databases until 16 September 2022, for all manuscripts reporting HSCT applications in *CSF1R*-ALSP and found data on 13 patients treated with HSCT [4,6,7,8,9,10,11]. We were able to follow up on 5 of these patients. Additionally, we retrieved data on 2 new unreported cases. Thus, a total of 15 patients (12 from the United States, 1 from Poland, 1 from France, and 1 from Norway) with *CSF1R*-ALSP treated with HSCT were included in the study. Data extraction and retrospective analysis were performed in a structured manner.

Information was collected regarding demographics (age of onset, sex, age of HSCT, disease duration before HSCT, post-HSCT follow-up duration), initial symptoms, pre-HSCT predominant symptomatology, pre-HSCT activities of daily living (ADLs), pre-HSCT professional activity, pre-HSCT clinical symptoms (cognitive decline, neuropsychiatric symptoms, extrapyramidal symptoms, pyramidal signs, gait problems, and seizures), clinical asymmetry, pre-HSCT brain magnetic resonance imaging (MRI) characteristics (white matter lesions (WMLs), cerebral atrophy, corpus callosum involvement, asymmetry), post-HSCT ADLs, post-HSCT professional activity, post-HSCT clinical outcomes (cognition, neuropsychiatric symptoms, extrapyramidal symptoms, pyramidal signs, gait problems, seizures), and post-HSCT radiological outcomes. Additionally, we calculated two combined measures of the post-HSCT outcome as follows. First, as a dichotomous combined outcome measure, the outcome was considered to be “good” if the patient was assessed as having either no change or improvement for each of the six symptom-related outcomes (allowing for missing data) or independent for ADLs or active for professional activity; all other patients were considered to have a “bad” outcome. Second, we calculated a score for each patient by summing the values for the individual post-HSCT outcomes (Symptoms: 0 = deterioration, 1 = no change, 2 = improvement; ADLs: 0 = dependent, 2 = independent; Professional activity: 0 = not active, 2 = active), where missing data for a post-HSCT outcome was imputed by using the mean value of the patients who did have data for the given outcome in order to avoid scores that are biased too low due to the presence of missing data. Therefore, a higher score indicates better post-HSCT outcomes.

## 3. Statistical Analysis

Continuous variables were summarized with the sample median and range. Categorical variables were summarized with numbers and percentages. Comparisons of demographics and pre-HSCT characteristics according to outcomes were made using a van Elteren stratified Wilcoxon rank sum test (continuous variables), or a Cochran-Mantel-Haenszel exact test (categorical variables), where the tests were stratified by median follow-up length after HSCT (≤26 months or >26 months). For outcomes that were measured as three-level categorical variables of either deterioration, no change, or improvement, the no change and improvement categories were combined and compared vs. the deterioration category in statistical analysis. Only outcomes that had a minimum of five patients in both categories were examined in statistical comparisons, and only demographics and pre-HSCT characteristics that were measured in at least 10 of the 15 patients were included in these comparisons. Associations of demographics and pre-HSCT characteristics with the combined outcome score were assessed using linear regression models that were adjusted for follow-up length after HSCT as a continuous variable. No adjustment for multiple testing was made in these exploratory analyses, and *p*-values < 0.05 were considered as statistically significant. All statistical tests were two-sided. Statistical analyses were performed using SAS (version 9.4; SAS Institute, Inc., Cary, NC, USA).

## 4. Results

Fifteen patients, 14 symptomatic and 1 asymptomatic, with *CSF1R*-ALSP treated with HSCT were included in the study. All 15 patients harbored different *CSF1R* mutations (Table 1). Patients’ demographics and pre-HSCT characteristics are summarized in Table 2. The median age of onset was 39 years (range 32–50 years; mean of 39.5 years), 4 patients (26.7%) were male, and the median age of HSCT was 43 years (range 31–52 years). Initial symptoms were cognitive impairment (42.9%), gait problems (21.4%), neuropsychiatric symptoms (21.4%), and other (14.3%). The median length of time from HSCT to the last follow-up evaluation was 26 months (range 3–180 months), including one patient who died 88 days after HSCT. Regarding post-HSCT outcomes (Table 3), an assessment of either no change or improvement occurred in 60.0% of patients for cognition, in 66.7% for neuropsychiatric symptoms, in 53.8% for extrapyramidal symptoms, in 75.0% for pyramidal signs, in 73.3% for gait problems, and in 90.9% for seizures. Post-HSCT ADLs were dependent for 54.5% of patients, and post-HSCT professional activity was not active for 66.7% of patients. For the combined outcome measures, 6 patients (40.0%) had a good outcome, and the median combined outcome score was 7 (Range: 1–14).

Of the individual post-HSCT outcomes, only cognition, neuropsychiatric symptoms, extrapyramidal symptoms, and ADLs had a minimum of 5 patients in both categories and were therefore examined in statistical analysis. Comparisons of demographics and pre-HSCT characteristics according to these outcomes are shown in Table 4 (cognition, neuropsychiatric symptoms) and Table 5 (extrapyramidal symptoms, activities of daily living). Patients with no change or improvement regarding post-HSCT cognition were less likely to have cognitive impairment as their initial symptom (12.5% vs. 83.3%, *p* = 0.025) and were more likely to have gait problems as their predominant symptom (80.0% vs. 0.0%, *p* = 0.033) compared to patients with deterioration on cognition (Table 4). Additionally, patients independent in post-HSCT ADLs were less likely to have pre-HSCT pyramidal signs compared to patients who were dependent (60.0% vs. 100.0%, *p* = 0.048, Table 5). No other statistically significant differences were noted regarding individual post-HSCT outcomes, though several suggestive (*p* < 0.10) findings were observed (Table 4 and Table 5).

Associations of demographics and pre-HSCT symptoms with combined outcome measures are displayed in Table 6. For the dichotomous combined outcome measure, compared to patients with a bad outcome, patients with a good outcome less often had an initial symptom of cognitive impairment (0.0% vs. 66.7%, *p* = 0.016), more often had an initial symptom of gait problems (60.0% vs. 0.0%, *p* = 0.041), had a younger age at HSCT (Median: 38 vs. 45 years, *p* = 0.044), and more often had a predominant symptom of gait problems (100.0% vs. 0.0%, *p* = 0.017). Regarding the combined outcome score, this was higher (indicating better outcomes) for patients with an initial symptom of gait problems (*p* = 0.009) and for patients whose predominant symptom was gait problems (*p* = 0.001).

## 5. Discussion

We found that gait problems as the initial and predominant manifestation of the disease were the most consistent predictors of good treatment outcomes. Moreover, younger age and an absence of pyramidal signs before HSCT increased the likelihood of good outcomes. There was also a suggestive association between disease duration before HSCT and treatment outcomes, with the shorter duration having better post-HSCT results, but it did not reach statistical significance. Conversely, prominent cognitive dysfunction was the most consistent predictor of bad outcomes following HSCT. The strongest link was observed for cognitive impairment as the first manifestation; however, there was also a suggestive association with cognitive impairment as the predominant symptom.

We found a significant link between the initial presentation with cognitive symptoms and the worsening thereof after HSCT. There was also a suggestive association between cognitive decline as a predominant symptom before HSCT and deterioration thereof after the treatment. There are no studies on quality of life (QoL) in *CSF1R*-ALSP; however, in other neurodegenerative diseases, cognition was shown to be one of the most important determinants of the QoL of patients and their caregivers [12,13,14]. Therefore, the HSCT’s positive impact on motor symptoms may be counterbalanced or even thwarted in case of cognitive deterioration. As the patients with a higher burden of cognitive symptoms were the most likely to suffer further decline following HSCT, and cognitive dysfunction was an independent predictor of bad outcomes following HSCT, patients with prominent pre-HSCT cognitive symptoms seem to be bad candidates for HSCT.

The higher-than-expected rate of bad outcomes (60%) in the present study is disconcerting. However, there are no previous studies systemically analyzing the HSCT outcomes in *CSF1R*-ALSP to compare the results. It may be the case that the HSCT group had a more aggressive form of the disease with a worse prognosis than other *CSF1R*-ALSP patients. This may be supported by the lower mean age of onset in our cohort (39.5 years) than the previously estimated (43 years) in the group of 122 patients with *CSF1R*-ALSP [15]. Contrarily, the first manifestation of cognitive impairment and general incidence of thereof was higher in the previously reported patients (59% and 94%, respectively) than in our cohort (43% and 80%, respectively) [15]. The prevalence of pyramidal signs was similar in the previously reported group (81%) and ours (79%) [15]. Comparing other motor symptoms between our cohort and previously reported patients is difficult, as different symptom classifications were used [1,2,15]. Parkinsonism, other involuntary movements, and ataxia were present in 61%, 21%, and 27% of the previously reported group of 122 *CSF1R*-ALSP patients [15]. In our group, 77% displayed extrapyramidal symptoms, and 73% had gait problems. Gait problems as the primary manifestation in some patients were recognized only recently, and the previous study did not consider them separately [1,2,15]. As gait abnormality may result from various motor symptoms, often coexisting, we think it is a particularly good measure of the motor-predominant phenotype. In the previous study, 19% developed motor symptoms as their first manifestation compared to 21% in the previous group [15]. Thus, our group generally did not differ significantly from other reported patients with *CSF1R*-ALSP and could be considered a representative sample.

On the other hand, the bad outcome in most patients in the present study may be attributed to applying very strict criteria for good outcomes following HSCT. The reported pre-HSCT high rate of ADLs dependence and low rate of professional activity (46% and 57%, respectively), that only slightly deteriorated during post-HSCT follow-up (55% and 67%, respectively), together with observed in most patients no change or improvement in clinical symptoms post-HSCT, suggest at least stabilization of the disease in the majority of patients. Furthermore, given the aggressive natural course of the disease leading inevitably to death within few years from the onset, we think most patients actually benefited from the treatment.

This study also provides evidence of at least two different subtypes of *CSF1R*-ALSP, a milder one manifesting primarily with motor phenotype and another more severe with predominant cognitive dysfunction. These subtypes not only differ by clinical manifestations but, most importantly, respond differently to HSCT and thus have distinct prognoses. As the treated group was small, it was impossible to investigate the two subtypes in more detail. The previous studies did not identify genotype-phenotype correlations [1,15]. Neuroimaging studies provided evidence of the volume of WMLs correlating with cognitive dysfunction [16]. Another study showed correlations between the extent of axonal loss on neuropathological examination and global disability and cognition [17,18]. However, *CSF1R*-ALSP subtypes per se and their radiopathological characteristics were not elucidated.

Understanding the molecular and pathological basis underlying phenotypic dichotomy in *CSF1R*-ALSP is challenging yet warrants further investigation. *CSF1R*-ALSP is a very rare disease, with approximately 300 cases reported worldwide. To further confound the picture, cognitive dysfunction and gait disturbances are both heterogenous groups of disorders with various pathological underpinnings. Lastly, the basis of these symptoms is not fully understood even in other, more common disorders, like multiple system atrophy, pseudotumor cerebri, or stroke [19,20,21]. In a study assessing outcomes of HSCT in adult cerebral X-linked adrenoleukodystrophy, the limited motor dysfunction was predictive of better treatment results [22]. Thus, despite some similarities between the disorders, their pathophysiological foundations and treatments must be considered separately.

The main limitation of this study is the small sample size, which results in a lack of power to detect associations with outcomes. Therefore, the possibility of a type II error (i.e., a false-negative finding) is important to consider, and we cannot conclude that no true association exists simply due to the occurrence of a non-significant *p*-value in this study. The limited sample size also prevents a rigorous multivariable analysis assessing independent predictors of outcome following HSCT, or development of an algorithm or scoring system that can be used to predict outcome. With more patients undergoing HSCT in the future, it might be possible to compare the two distinct phenotypes regarding the age of disease onset, gender, and other important clinical variables to further clarify our observations.

## 6. Conclusions

Two distinct phenotypes of *CSF1R*-ALSP with different clinical presentations and responses to HSCT were recognized. Gait problems as initial manifestation and predominant symptomatology before HSCT indicated a milder disease phenotype with better response to HSCT and good therapy outcomes. On the opposite end were the patients who developed cognitive symptoms first, as they were the least likely to benefit from HSCT.

## Figures and Tables

**Table 1 pharmaceutics-14-02778-t001:** List of CSF1R mutations of the 15 patients included into the study.

Mutation	No. (%) of Patients (N = 15)
c.2357 T > C	1 (6.7%)
aberrant splice variant c.2442+1G > A	1 (6.7%)
c.1754-2A > G	1 (6.7%)
c.1924 C > T	1 (6.7%)
c.1996 C > T	1 (6.7%)
c.2345 G > A	1 (6.7%)
c.2375 C > A	1 (6.7%)
c.2381T > C	1 (6.7%)
c.2392G > C	1 (6.7%)
c.2450 T > A	1 (6.7%)
c.2498 C > A	1 (6.7%)
c.2624 T > C	1 (6.7%)
c.2625 G > A	1 (6.7%)
c.2677 T > C	1 (6.7%)
c.1990 G > A	1 (6.7%)

**Table 2 pharmaceutics-14-02778-t002:** Summary of patients’ demographics and pre-HSCT characteristics.

Variable	N	Median (Minimum, Maximum) or No. (%) of Patients
Age of onset (years)	14	39 (32, 50); mean = 39.5
Sex (Male)	15	4 (26.7%)
Initial symptom	14	
Cognitive impairment		6 (42.9%)
Gait problems		3 (21.4%)
Neuropsychiatric symptoms		3 (21.4%)
Other		2 (14.3%)
Age of HSCT (years)	15	43 (31, 52)
Length of time from disease onset to HSCT (months)	14	24 (11, 144)
Pre-HSCT cognitive decline	15	12 (80.0%)
Pre-HSCT neuropsychiatric symptoms	15	11 (73.3%)
Pre-HSCT extrapyramidal symptoms	13	10 (76.9%)
Pre-HSCT pyramidal signs	14	11 (78.6%)
Pre-HSCT gait problems	15	11 (73.3%)
Pre-HSCT seizures	13	13 (100.0%)
Pre-HSCT predominant symptom	11	
Cognitive impairment		4 (36.4%)
Gait problems		4 (36.4%)
Neuropsychiatric symptoms		2 (18.2%)
Other		1 (9.1%)
Pre-HSCT activities of daily living (dependent)	11	5 (45.5%)
Pre-HSCT professional activity (not active)	7	4 (57.1%)
Pre-HSCT clinical asymmetry	10	6 (60.0%)
Pre-HSCT imaging white matter lesions	15	14 (93.3%)
Pre-HSCT imaging atrophy	15	14 (93.3%)
Pre-HSCT corpus callosum involvement	9	8 (88.9%)
Pre-HSCT imaging asymmetry	11	6 (54.5%)

HSCT—hematopoietic stem cell transplantation.

**Table 3 pharmaceutics-14-02778-t003:** Summary of post-HSCT outcomes.

Variable	N	Median (Minimum, Maximum) or No. (%) of Patients
Length of time from HSCT to follow-up evaluation (months)	15	26 (3, 180)
Post-HSCT cognition	15	
Deterioration		6 (40.0%)
No change		6 (40.0%)
Improvement		3 (20.0%)
Post-HSCT neuropsychiatric symptoms	15	
Deterioration		5 (33.3%)
No change		6 (40.0%)
Improvement		4 (26.7%)
Post-HSCT extrapyramidal symptoms	13	
Deterioration		6 (46.2%)
No change		7 (53.8%)
Improvement		0 (0.0%)
Post-HSCT pyramidal signs	12	
Deterioration		3 (25.0%)
No change		7 (58.3%)
Improvement		2 (16.7%)
Post-HSCT gait problems	15	
Deterioration		4 (26.7%)
No change		8 (53.3%)
Improvement		3 (20.0%)
Post-HSCT seizures	11	
Deterioration		1 (9.1%)
No change		10 (90.9%)
Improvement		0 (0.0%)
Post-HSCT activities of daily living (dependent)	11	6 (54.5%)
Post-HSCT professional activity (not active)	9	6 (66.7%)
Dichotomous combined outcome	15	
Good outcome		6 (40.0%)
Bad outcome		9 (60.0%)
Combined outcome score	15	7 (1, 14)

HSCT—hematopoietic stem cell transplantation.

**Table 4 pharmaceutics-14-02778-t004:** Associations of patient characteristics with post-HSCT cognition and neuropsychiatric symptoms.

	Post-HSCT Cognition			Post-HSCT Neuropsychiatric Symptoms		
	Median (Minimum, Maximum) or No. (%) of Patients			Median (Minimum, Maximum) or No. (%) of Patients		
Variable	Deterioration (N = 6)	No Change or Improvement (N = 9)	*p*-Value	Deterioration (N = 5)	No Change or Improvement (N = 10)	*p*-Value
Age of onset (years)	42 (36, 50)	37 (32, 43)	0.19	42 (36, 50)	37 (32, 43)	0.25
Sex (Male)	2 (33.3%)	2 (22.2%)	1.00	2 (40.0%)	2 (20.0%)	1.00
Initial symptom						
Cognitive impairment	5 (83.3%)	1 (12.5%)	0.025	4 (80.0%)	2 (22.2%)	0.065
Gait problems	0 (0.0%)	3 (37.5%)	0.23	0 (0.0%)	3 (33.3%)	0.47
Neuropsychiatric symptoms	1 (16.7%)	2 (25.0%)	1.00	1 (20.0%)	2 (22.2%)	1.00
Other	0 (0.0%)	2 (25.0%)	0.47	0 (0.0%)	2 (22.2%)	0.45
Age of HSCT (years)	45 (41, 51)	39 (31, 52)	0.28	44 (41, 51)	40 (31, 52)	0.44
Length of time from disease onset to HSCT (months)	30 (11, 60)	24 (18, 144)	0.89	24 (11, 60)	24 (18, 144)	0.65
Pre-HSCT cognitive decline	6 (100.0%)	6 (66.7%)	0.21	5 (100.0%)	7 (70.0%)	0.21
Pre-HSCT neuropsychiatric symptoms	5 (83.3%)	6 (66.7%)	0.57	4 (80.0%)	7 (70.0%)	0.57
Pre-HSCT extrapyramidal symptoms	5 (83.3%)	5 (71.4%)	1.00	5 (100.0%)	5 (62.5%)	0.50
Pre-HSCT pyramidal signs	5 (83.3%)	6 (75.0%)	1.00	5 (100.0%)	6 (66.7%)	0.21
Pre-HSCT gait problems	3 (50.0%)	8 (88.9%)	0.24	3 (60.0%)	8 (80.0%)	1.00
Pre-HSCT seizures	0 (0.0%)	0 (0.0%)	1.00	0 (0.0%)	0 (0.0%)	1.00
Pre-HSCT predominant symptom						
Cognitive impairment	4 (66.7%)	0 (0.0%)	0.060	3 (60.0%)	1 (16.7%)	0.20
Gait problems	0 (0.0%)	4 (80.0%)	0.033	0 (0.0%)	4 (66.7%)	0.13
Neuropsychiatric symptoms	2 (33.3%)	0 (0.0%)	0.47	2 (40.0%)	0 (0.0%)	0.47
Other	0 (0.0%)	1 (20.0%)	0.33	0 (0.0%)	1 (16.7%)	0.33
Pre-HSCT activities of daily living (dependent)	3 (60.0%)	2 (33.3%)	0.57	3 (60.0%)	2 (33.3%)	0.57
Pre-HSCT clinical asymmetry	2 (50.0%)	4 (66.7%)	1.00	2 (50.0%)	4 (66.7%)	1.00
Pre-HSCT imaging white matter lesions	6 (100.0%)	8 (88.9%)	1.00	5 (100.0%)	9 (90.0%)	1.00
Pre-HSCT imaging atrophy	6 (100.0%)	8 (88.9%)	1.00	5 (100.0%)	9 (90.0%)	1.00
Pre-HSCT imaging asymmetry	1 (25.0%)	5 (71.4%)	0.26	1 (25.0%)	5 (71.4%)	0.26

*p*-values result from a van Elteren stratified Wilcoxon rank sum test (continuous variables), or a Cochran-Mantel-Haenszel exact test, where the tests were stratified by median follow-up length after HSCT (≤26 months or >26 months). HSCT—hematopoietic stem cell transplantation.

**Table 5 pharmaceutics-14-02778-t005:** Associations of patient characteristics with post-HSCT cognition and neuropsychiatric symptoms.

	Post-HSCT Extrapyramidal Symptoms			Post-HSCT Activities of Daily Living		
	Median (Minimum, Maximum) or No. (%) of Patients			Median (Minimum, Maximum) or No. (%) of Patients		
Variable	Deterioration (N = 6)	No Change or Improvement (N = 9)	*p*-Value	Dependent (N = 5)	Independent (N = 10)	*p*-Value
Age of onset (years)	43 (36, 50)	39 (35, 43)	0.27	41 (36, 50)	36 (32, 43)	0.80
Sex (Male)	2 (33.3%)	2 (28.6%)	1.00	2 (33.3%)	1 (20.0%)	1.00
Initial symptom						
Cognitive impairment	4 (66.7%)	2 (33.3%)	0.31	4 (66.7%)	0 (0.0%)	0.13
Gait problems	0 (0.0%)	3 (50.0%)	0.19	0 (0.0%)	3 (75.0%)	0.083
Neuropsychiatric symptoms	1 (16.7%)	1 (16.7%)	1.00	1 (16.7%)	0 (0.0%)	1.00
Other	1 (16.7%)	0 (0.0%)	1.00	1 (16.7%)	1 (25.0%)	1.00
Age of HSCT (years)	45 (41, 52)	43 (31, 49)	0.38	44 (41, 51)	38 (31, 45)	0.28
Length of time from disease onset to HSCT (months)	24 (11, 108)	30 (24, 144)	0.63	24 (11, 60)	24 (18, 24)	0.55
Pre-HSCT cognitive decline	5 (83.3%)	6 (85.7%)	1.00	5 (83.3%)	3 (60.0%)	0.52
Pre-HSCT neuropsychiatric symptoms	4 (66.7%)	5 (71.4%)	1.00	4 (66.7%)	3 (60.0%)	1.00
Pre-HSCT extrapyramidal symptoms	6 (100.0%)	4 (57.1%)	0.20	6 (100.0%)	2 (50.0%)	0.19
Pre-HSCT pyramidal signs	6 (100.0%)	4 (57.1%)	0.21	6 (100.0%)	3 (60.0%)	0.048
Pre-HSCT gait problems	5 (83.3%)	4 (57.1%)	0.57	4 (66.7%)	4 (80.0%)	1.00
Pre-HSCT seizures	6 (100.0%)	7 (100.0%)	1.00	0 (0.0%)	0 (0.0%)	1.00
Pre-HSCT predominant symptom						
Cognitive impairment	2 (40.0%)	2 (40.0%)	1.00	3 (50.0%)	0 (0.0%)	0.44
Gait problems	0 (0.0%)	3 (60.0%)	0.17	0 (0.0%)	3 (100.0%)	0.056
Neuropsychiatric symptoms	2 (40.0%)	0 (0.0%)	0.47	2 (33.3%)	0 (0.0%)	1.00
Other	1 (20.0%)	0 (0.0%)	1.00	1 (16.7%)	0 (0.0%)	1.00
Pre-HSCT activities of daily living (dependent)	4 (80.0%)	0 (0.0%)	0.057	4 (66.7%)	1 (20.0%)	0.21
Pre-HSCT clinical asymmetry	4 (80.0%)	2 (40.0%)	0.53	3 (60.0%)	1 (33.3%)	1.00
Pre-HSCT imaging white matter lesions	6 (100.0%)	6 (85.7%)	1.00	6 (100.0%)	4 (80.0%)	0.29
Pre-HSCT imaging atrophy	6 (100.0%)	6 (85.7%)	1.00	6 (100.0%)	4 (80.0%)	0.29
Pre-HSCT imaging asymmetry	3 (60.0%)	2 (40.0%)	1.00	2 (40.0%)	2 (50.0%)	1.00

*p*-values result from a van Elteren stratified Wilcoxon rank sum test (continuous variables), or a Cochran-Mantel-Haenszel exact test, where the tests were stratified by median follow-up length after HSCT (≤26 months or >26 months). HSCT—hematopoietic stem cell transplantation.

**Table 6 pharmaceutics-14-02778-t006:** Associations of patient characteristics with post-HSCT combined outcome measures.

	Dichotomous Combined Outcome			Combined Outcome Score		
	Median (Minimum, Maximum) or No. (%) of Patients			Median (Minimum, Maximum) or No. (%) of Patients		
Variable	Bad Outcome (N = 9)	Good Outcome (N = 6)	*p*-Value	Score ≤ 7 (N = 8)	Score > 7 (N = 7)	*p*-Value
Age of onset (years)	42 (36, 50)	35 (32, 43)	0.25	41 (35, 50)	37 (32, 43)	0.16
Sex (Male)	3 (33.3%)	1 (16.7%)	1.00	2 (25.0%)	2 (28.6%)	0.75
Initial symptom						
Cognitive impairment	6 (66.7%)	0 (0.0%)	0.016	5 (62.5%)	1 (16.7%)	0.070
Gait problems	0 (0.0%)	3 (60.0%)	0.041	0 (0.0%)	3 (50.0%)	0.009
Neuropsychiatric symptoms	2 (22.2%)	1 (20.0%)	1.00	2 (25.0%)	1 (16.7%)	0.60
Other	1 (11.1%)	1 (20.0%)	1.00	1 (12.5%)	1 (16.7%)	0.80
Age of HSCT (years)	45 (41, 52)	38 (31, 45)	0.044	44 (37, 52)	39 (31, 49)	0.26
Length of time from disease onset to HSCT (months)	36 (11, 144)	24 (18, 24)	0.086	24 (11, 108)	24 (18, 144)	0.76
Pre-HSCT cognitive decline	8 (88.9%)	4 (66.7%)	0.51	7 (87.5%)	5 (71.4%)	0.62
Pre-HSCT neuropsychiatric symptoms	7 (77.8%)	4 (66.7%)	0.57	6 (75.0%)	5 (71.4%)	0.78
Pre-HSCT extrapyramidal symptoms	8 (88.9%)	2 (50.0%)	0.48	7 (100.0%)	3 (50.0%)	0.10
Pre-HSCT pyramidal signs	8 (88.9%)	3 (60.0%)	0.25	7 (100.0%)	4 (57.1%)	0.14
Pre-HSCT gait problems	6 (66.7%)	5 (83.3%)	1.00	6 (75.0%)	5 (71.4%)	0.62
Pre-HSCT seizures	0 (0.0%)	0 (0.0%)	N/A	0 (0.0%)	0 (0.0%)	N/A
Pre-HSCT predominant symptom						
Cognitive impairment	4 (57.1%)	0 (0.0%)	0.067	3 (42.9%)	1 (25.0%)	0.37
Gait problems	0 (0.0%)	4 (100.0%)	0.017	1 (14.3%)	3 (75.0%)	0.001
Neuropsychiatric symptoms	2 (28.6%)	0 (0.0%)	1.00	2 (28.6%)	0 (0.0%)	0.18
Other	1 (14.3%)	0 (0.0%)	1.00	1 (14.3%)	0 (0.0%)	0.74
Pre-HSCT activities of daily living (dependent)	4 (66.7%)	1 (20.0%)	0.21	4 (66.7%)	1 (20.0%)	0.16
Pre-HSCT clinical asymmetry	5 (71.4%)	1 (33.3%)	0.50	4 (66.7%)	2 (50.0%)	0.80
Pre-HSCT imaging white matter lesions	9 (100.0%)	5 (83.3%)	0.25	8 (100.0%)	6 (85.7%)	0.60
Pre-HSCT imaging atrophy	9 (100.0%)	5 (83.3%)	0.25	8 (100.0%)	6 (85.7%)	0.60
Pre-HSCT imaging asymmetry	4 (57.1%)	2 (50.0%)	1.00	3 (50.0%)	3 (60.0%)	0.35

For analysis of the dichotomous combined outcome, *p*-values result from a van Elteren stratified Wilcoxon rank sum test (continuous variables), or a Cochran-Mantel-Haenszel exact test, where the tests were stratified by median follow-up length after HSCT (≤26 months or >26 months). For analysis of the combined outcome score, *p*-values results from linear regression models that were adjusted for follow-up length after HSCT (where follow-up length is considered as a continuous variable); the combined outcome score was assessed as a continuous variable in linear regression analysis, and was dichotomized based on the sample median (≤7 vs. >7) for purposes of presentation only. HSCT—hematopoietic stem cell transplantation.

## Data Availability

Data (including a detailed description of individual cases) available on request from the corresponding author.

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
