# Peer review of "Hematopoietic Stem Cell Transplantation in CSF1R-Related Leukoencephalopathy: Retrospective Study on Predictors of Outcomes"

_pharmaceutics, 2022, doi:10.3390/pharmaceutics14122778_

Round 1

Reviewer 1 Report

The authors have studied patients with CSF1R-related leukoencephalopathy after HSCT. They found two distinct phenotypes of CSF1R-ALSP with different clinical presentations and different clinical response. Comments:

1. Please, check age at onset and age at HSCT, the lower range is lower at HSCT compared to at onset!

2. No clear conclusions can be drawn from a study of this size. It would however be interesting if the authors could compare the two distinct phenotypes concerning age, gender, etc.  

Author Response

Reviewer1:

The authors have studied patients with CSF1R-related leukoencephalopathy after HSCT. They found two distinct phenotypes of CSF1R-ALSP with different clinical presentations and different clinical response. Comments:

REPLY: Many thanks for your thorough review and thoughtful comments on our work. In the following, we give a point-by-point reply to your comments.

  1. Please, check age at onset and age at HSCT, the lower range is lower at HSCT compared to at onset!

REPLY: The apparent discrepancy between the ranges of age at HSCT and age of onset, is not a mistake. It results from the fact that one patient was asymptomatic and was not included in the "age of onset" calculations, but she was included in the "age of HSCT". We added the following information to the manuscript to make it clearer:

Abstract:

We retrospectively analyzed 15 patients, 14 symptomatic and 1 asymptomatic, with CSF1R-ALSP that underwent HSCT.

Results:

Fifteen patients with CSF1R-ALSP, 14 symptomatic and 1 asymptomatic, treated with HSCT were included in the study.

  1. No clear conclusions can be drawn from a study of this size. It would however be interesting if the authors could compare the two distinct phenotypes concerning age, gender, etc.

REPLY: We would be happy to do so; however, completing the work within the next few days as requested by the Editor-in-Chief would be very difficult. We are in midst of US national holiday called Thanksgiving and our statistician (Dr. Michael Heckman) is on vacation. Therefore, we would need much more time to conduct an analysis and add it to the manuscript. However, the group with good outcomes mainly consisted of patients with gait problems as the predominant phenotype, whereas the group with the bad results was mostly composed of patients with predominant cognitive impairment. Hence, these two phenotypes were already indirectly analyzed and compared with each other. In addition, the two groups consist of only a few patients each and it would be difficult to draw any results with enough statistical power. Therefore it is unlikely that the new calculations will add much to the paper. Therefore, we have not changed our manuscript. Nevertheless, we have added a statement to the limitation part of our manuscript on this point that reads as follows:

With more patients undergoing HSCT in the future, it might be possible to compare the two distinct phenotypes regarding the age of disease onset, gender, and other important clinical variables to further clarify our observations.

Reviewer 2 Report

This manuscript describes valuable information regarding the predictors of effective treatment for relatively rare disorders. 

This reviewer has no objection to publish this manuscript.

Author Response

Reviewer 2:

This manuscript describes valuable information regarding the predictors of effective treatment for relatively rare disorders.

This reviewer has no objection to publish this manuscript.

REPLY: Thank you very much for your review and positive feedback.